# PRIMARY-DUAL DIFFUSION SOLVER FOR QUADRATIC PROGRAMMING PROBLEMS

## ABSTRACT

Quadratic Programming (QP) is an important class of mathematical optimization problems widely used in various fields such as economics, engineering, finance, and machine learning. Recently, with the development of Learning to Optimize , many studies have attempted to solve QP problems using Graph Neural Networks (GNNs), but they suffer from relatively poor performance compared to traditional algorithms. In this paper, we introduce the Primary-Dual Diffusion (PDD) model for solving QP problems. The model uses a diffusion approach to simultaneously learn both primary and dual variables in order to predict an accurate solution. Based on this prediction, only a small number of KKT-based correction and parallelizable post-processing iterations (e.g, PDHG, ADMM) are needed to ensure that the solution satisfies the constraints and converges to the optimal solution. Notably, our PDDQP is the first QP neural solver capable of obtaining the optimal solution. Additionally, to address the slow convergence issue of the diffusion model, we adopt a consistency distillation method to develop a one-step diffusion solver for QP. Experimental results demonstrate that our approach achieves state-of-the-art performance in learning-based QP solvers while remaining competitive with traditional methods.

## 1 INTRODUCTION

Quadratic Programming (QP) stands as a fundamental class of optimization problems, finding widespread application across diverse fields, including robotics and control Bemporad et al. (2002); Kouzoubov et al. (2019). In finance Markowitz (1952), in machine learning Cortes & Vapnik (1995);Szeliski (2006); Shrivastava & Szeliski (2015), in power systems Zhang & Xiao (2017).Its ability to model a convex quadratic objective function subject to linear constraints makes it a powerful tool for solving problems that balance competing goals while adhering to physical or logical limits. Beyond these direct applications, QP also plays a crucial role as a building block for solving more complex, non-linear optimization challenges. Many advanced algorithms, such as sequential quadratic programming Han (1977) and trust-region methods Moré (1983), repeatedly solve a series of simplified QP subproblems to find a solution to the original, more difficult problem.

Current solving frameworks for quadratic programming problems face two major challenges. On the one hand, traditional optimization algorithms, such as interior-point methods Karmarkar (1984); Nocedal & Wright (2006) and first-order methods, possess solid theoretical convergence guarantees. However, their application is often hindered by high computational costs or slow convergence rates. For instance, the computational complexity of matrix decomposition in interior-point methods can be as high as $O(n^3)$, while first-order methods may require thousands of iterations to achieve high accuracy, making them unsuitable for many real-time applications. On the other hand, data-driven methods represented by Graph Neural Networks (GNNs) have emerged to improve solving efficiency Chen et al. (2024); Wu et al. (2024). Nevertheless, when these purely supervised learning paradigms are used to directly predict the primal solution, they exhibit significant shortcomings: firstly, the predictive accuracy is limited, making it difficult to approximate the optimal solution for complex problems; secondly, the lack of theoretical guarantees means the predicted solutions often violate problem constraints. Consequently, relying solely on end-to-end prediction cannot ensure solution quality and feasibility. To overcome these deficiencies, it is crucial to integrate the fundamental properties of optimization problems—such as the primal-dual variables and KKT con-

ditions—as physical constraints into the model training process. This guides the model to generate solutions that are both efficient and theoretically sound.

**Contributions.** This paper proposes a novel diffusion model-based quadratic programming (QP) solving framework called **PDDQP** (and its refined variant PDDQP-R), which deeply integrates deep learning with classical optimization methods to achieve an effective balance between computational efficiency and solution accuracy. The main contributions are as follows:

(1) Unlike previous approaches that only learn primal variables, we propose a **joint learning method for primal and dual variables** to improve prediction accuracy. Our approach aligns with the structure of QP problems, which can be represented as bipartite graphs. We introduce a message-passing graph neural network (GNN) that simultaneously processes information from variable nodes and constraint nodes, enabling synchronous and high-accuracy prediction of both primal and dual variables.

(2) We apply diffusion models to QP and obtain a single-step solver through consistency distillation. We argue that this distillation paradigm is among the most suitable diffusion learning frameworks for QP tasks. According to our experiments, other efficient inference generative models, such as self-trained consistency models based on ODEs and flow matching-based one-step generators, have proven inadequate for quadratic programming, which is highly sensitive to numerical accuracy. In contrast, our method samples point pairs along the diffusion trajectory and directly minimizes the output discrepancy between student and teacher models at the denoised solution. This provides strong supervision for the student's solution mapping, demonstrating an effective way to compress the multi-step process into a single-step sampler while retaining high-fidelity numerical priors. Additionally, during the sampling process of the teacher model, we incorporated additional gradient guidance using the KKT conditions. This approach enables the generation of primal-dual variables that more closely adhere to the KKT conditions, particularly the stationarity condition.

(3) We have explored and proposed several post-refinement processing methods that integrate traditional algorithms. When addressing convex optimization problems with linear constraints such as QP, conventional optimization algorithms often involve simultaneous updates of primal and dual variables (e.g., ADMM, PDHG), which aligns closely with our objective of concurrently outputting both primal and dual variables. The output of our diffusion model provides a favorable starting point for these optimization methods. Empirical results demonstrate that, building upon the diffusion-based learning process, only a few steps of parallel post-processing are sufficient to obtain a convergent, constraint-satisfying, and near-optimal solution. As the problem dimension increases, industrial-grade solvers such as Gurobi exhibit significant computational slowdown, whereas our diffusion-based approach demonstrates growing advantages in this regime.explored and proposed some post-refinement processing methods, including utilizing KKT conditions to optimize dual and primal variables, and employing traditional algorithms for iterative refinement. Empirically, by building upon the diffusion-based learning process, we show that only a few steps of **parallel post-processing** are sufficient to obtain a convergent, constraint-satisfying, and near-optimal solution. In our experiments, as the problem dimension increases, industrial-grade solvers such as Gurobi exhibit significant computational slowdown, whereas our diffusion-based approach demonstrates growing advantages in this regime.

## 2 PRELIMINARY AND RELATED WORK

A convex quadratic programming (QP) problem and its corresponding dual problem can be expressed as:

$$\min_{x \in \mathbb{R}^n} \frac{1}{2} x^\top Q x + c^\top x \quad \text{s.t.} \quad Ax = b, \ x \geq 0, \tag{1}$$

where $x \in \mathbb{R}^n$ is the primal decision vector, and the problem data is given by $\zeta = \{Q \in \mathbb{S}^n_{++}, c \in \mathbb{R}^n, A \in \mathbb{R}^{m \times n}, b \in \mathbb{R}^m\}$. For each primal QP, we introduce dual variables $y \in \mathbb{R}^m$ associated with the equality constraints $Ax = b$, and dual variables $s \geq 0$ associated with the non-negativity constraints $x \geq 0$. These dual variables measure the sensitivity of the optimal objective to perturbations in the constraints. For instance, $y$ indicates how the optimal value changes with modifications to $b$, while $s$ reflects the shadow price of enforcing the non-negativity constraints. And the dual problem

can be expressed as follows:

$$\max_{y\in\mathbb{R}^m, s\in\mathbb{R}^n} \quad -\frac{1}{2}(A^\top y + s - c)^\top Q^{-1}(A^\top y + s - c) + b^\top y \quad \text{s.t.} \quad s \geq 0. \tag{2}$$

At optimality, the primal variables $x^\star$ and the dual variables $(y^\star, s^\star)$ must satisfy the Karush-Kuhn-Tucker (KKT) conditions:

$$\begin{cases} Qx^\star + c - A^\top y^\star - s^\star = 0, & \text{(Stationarity)} \\ Ax^\star = b, & \text{(Primal Feasibility)} \\ x^\star \geq 0, \ s^\star \geq 0, & \text{(Primal and Dual Feasibility)} \\ x^\star \odot s^\star = 0. & \text{(Complementary Slackness)} \end{cases} \tag{3}$$

These conditions illustrate that the primal and dual solutions are interdependent and collectively characterize optimality. In particular, the complementary slackness condition ($x^\star \odot s^\star = 0$) directly links the primal variables to the dual variables of the non-negativity constraints. It implies that if a primal variable $x_i^\star$ is positive (i.e., the constraint $x_i \geq 0$ is not binding), then its corresponding dual variable $s_i^\star$ must be zero. This information is crucial for sensitivity analysis, constraint adjustments, and data-driven prediction methods.Traditional algorithms such as ADMM Boyd et al. (2011) and PDHG He et al. (2014) employ iterative optimization methods, simultaneously updating dual and primal variables while providing rigorous theoretical convergence guarantees. Specifically, ADMM and PDHG share the core concept of alternating updates, decomposing complex problems into simpler subproblems solved sequentially, making them naturally suitable for distributed computing environments . Interior-Point Methods (IPMs) are often regarded as state-of-the-art (SOTA) for solving medium-scale QP problems. By following the "central path" within the feasible region to converge to the optimal solution, IPMs offer theoretical guarantees of polynomial-time complexity Nesterov & Nemirovskii (1994). Despite their higher computational cost per iteration, their superlinear convergence rate makes them the preferred algorithm in many commercial solvers.However, despite their solid theoretical foundations, traditional algorithms exhibit significant limitations when dealing with ultra-large-scale problems, real-time requirements, or data-driven scenarios. ADMM and PDHG generally suffer from slow convergence rates, often requiring a large number of iterations to achieve high precision. Their performance is highly sensitive to parameter selection (e.g., step size, penalty parameters), necessitating extensive parameter tuning in practical applications. Although IPMs demonstrate fast convergence and high accuracy, each iteration requires the computation and storage of large-scale Hessian matrices or their approximations, along with solving linear systems. Their computational and memory complexity becomes prohibitive for ultra-high-dimensional problems.

**Machine Learning for Quadratic Programming.** Learning to Optimize has emerged as a prominent research focus in recent years. The pioneering work OptNet proposed by Amos et al. Amos & Kolter (2017) embeds constrained optimization problems such as quadratic programming (QP) as differentiable layers into neural networks, providing significant inspiration for subsequent research in this area.Algorithm unrolling represents a major category within learning to optimize. For instance, HEAP Feng et al. introduces a neural operator solver by reformulating constrained high-dimensional PDEs as QP problems and unrolling several steps of an adaptive primal–dual hybrid gradient (A-PDHG) algorithm. Through end-to-end learning of step sizes and momentum parameters, HEAP significantly reduces computational cost while maintaining accuracy and constraint satisfaction. Distributed learning-to-optimize methods have recently begun to demonstrate their advantages, though this direction remains largely unexplored. DeepDistributedQP Saravanos et al. (2024) proposes a distributed deep learning architecture that unrolls a novel DistributedQP algorithm into neural network layers to learn optimal parameter policies. This approach can scale to very high-dimensional problems, supported by PAC-Bayes McAllester (2013) generalization guarantees. Beyond learning-based optimization, another line of work employs machine learning methods—particularly graph neural networks (GNNs)—to produce end-to-end solutions. The study Expressive Power of Graph Neural Networks for (Mixed-Integer) Quadratic Programs Chen et al. (2024) rigorously analyzes the theoretical expressive power of GNNs in solving QP problems, opening new avenues for Quadratic Programming solving. Despite these promising advances, such methods still face notable limitations. Unrolled optimization networks often suffer from limited generalization beyond the distribution of training instances, and their stability can be sensitive to hyperparameter choices. GNN-based solvers, while expressive, struggle with scalability to very large-scale problems and often lack interpretability. Moreover, the theoretical understanding of the convergence

and robustness of learned optimizers—especially in constrained and distributed settings—remains incomplete. These challenges point to critical directions for future research in bridging machine learning and mathematical optimization.

**Diffusion-Based Models for Solving Optimization Problems.** In recent years, deep learning, and in particular generative models, have been increasingly applied to combinatorial and continuous optimization problems, including quadratic and linearly constrained programs. Diffusion models, as a powerful class of generative models, have recently demonstrated promising results in this context. The T2T (Training to Testing) framework Li et al. (2023) leverages a generative model during training to learn the distribution of high-quality solutions and employs a gradient-based search during testing to identify optimal solutions. This approach achieved significant performance improvements on combinatorial optimization tasks such as the Traveling Salesman Problem and Maximum Independent Set Problem. Similarly, DIFUSCO Sun & Yang (2023) introduces a graph-based diffusion framework for solving NP-complete combinatorial problems, surpassing prior state-of-the-art methods across multiple benchmarks. Zhao et al. (2024) introduced fundamental improvements targeting the core challenges of efficiency and sampling, leading to breakthroughs in both performance and computational speed. The success of T2T and DIFUSCO highlights the potential of diffusion-based methods to bring new breakthroughs to combinatorial and high-dimensional optimization, suggesting a promising direction for future research and practical applications. While diffusion architectures have demonstrated considerable potential in combinatorial optimization problems, to the best of our knowledge, their progress in planning problems—particularly continuous planning domains—remains limited. Our work fills this gap.

## 3 PRIMARY DUAL DIFFUSION FOR LEARNING QUADRATIC PROGRAMMING

### 3.1 JOINT PRIMARY AND DUAL VARIABLE LEARNING WITH DIFFUSION MODEL

This paper proposes an innovative one-step sampling diffusion model for efficiently solving quadratic programming (QP) problems citesong2023consistency. We have experimented with various training methods for diffusion models, such as self-training consistency models and Flow Matching-based consistency models Lipman et al. (2022). However, the results failed to match the superior performance of distillation-based techniques. According to the experimental results, the teacher diffusion model can find the optimal solution for the QP problem very accurately and stably through multi-step sampling. By integrating the strengths of diffusion models and consistency models, it transforms traditional optimization processes into a generative task.

The core of this consistency model involves training a Graph Neural Network (GNN) to perform reverse denoising on a combined solution vector $z = [x, y]$, which comprises the primal variables $x$ and dual variables $y$ Equation 2. During the training phase, the model receives a noisy vector $z_t$, generated from a ground-truth solution, along with the corresponding problem data, and outputs a prediction of the noise-free solution, $\hat{z}_0$, to compute the loss. At inference time, the model starts with a pure noise vector $z_T$ and the problem data, generating a high-quality initial solution $\hat{z}_0$ through one or more denoising steps. This initial solution can then be further refined by classical optimization algorithms like ADMM or PDHG to ultimately yield a final, optimized solution.

The framework first introduces a Denoising Diffusion Probabilistic Model (DDPM)Ho et al. (2020) as a teacher model. The forward diffusion process generates a noisy state $z_t$ by progressively adding Gaussian noise to the ground truth solution $z_0$. At any time step $t$, the distribution of $z_t$ can be expressed as:

$$q(z_t|z_0) = \mathcal{N}(z_t; \sqrt{\bar{\alpha}_t}z_0, (1 - \bar{\alpha}_t)I). \tag{4}$$

where $\bar{\alpha}_t$ is a predefined schedule parameter that controls the amount of noise added.

To facilitate the distillation process, our teacher model was trained to predict the solution directly. This is a valid reparameterization of the standard DDPM objective of predicting the noise $\epsilon$, as they are linked by the algebraic relation $z_0 = (z_t - \sqrt{1 - \bar{\alpha}_t}\epsilon)/\sqrt{\bar{\alpha}_t}$. And they are mathematically equivalent in form. The $z_0$-prediction paradigm is particularly suitable for consistency training. The network's prediction function is parameterized as:

$$\hat{z}_0 = f_\theta(\mathcal{G}, z_t, t). \tag{5}$$

where $\mathcal{G}$ represents the QP problem data formulated as a graph (including objective function and constraint parameters). For improved accuracy, the teacher model can employ a multi-step sampling

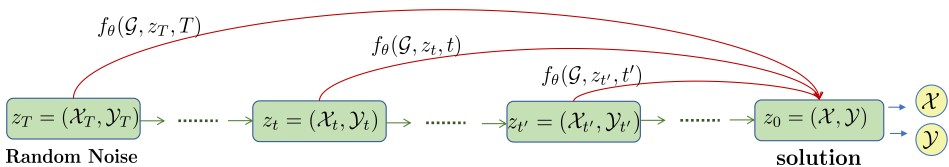

Figure 1: Illustration of our PDDQP. We first train a DDPM-like multi-step model for QP learning, and then apply consistency distillation to learn a one-step diffusion model for solving QP.

process. This process begins with a pure noise vector, $\boldsymbol{z}_T \sim \mathcal{N}(0, \boldsymbol{I})$, and progressively refines the solution. At each step $i$ (from $t_i$ to $t_{i-1}$), the model first predicts a clean solution $\hat{\boldsymbol{z}}_0$ from the current noisy state $\boldsymbol{z}_{t_i}$:

$$\hat{\boldsymbol{z}}_0 = f_\theta(\mathcal{G}, \boldsymbol{z}_{t_i}, t_i). \tag{6}$$

The noisy input for the next step, $\boldsymbol{z}_{t_{i-1}}$, is then computed by adding a controlled amount of new noise to the predicted clean solution:

$$\boldsymbol{z}_{t_{i-1}} = \sqrt{\bar{\alpha}_{t_{i-1}}}\hat{\boldsymbol{z}}_0 + \sqrt{1 - \bar{\alpha}_{t_{i-1}}}\boldsymbol{\epsilon}. \tag{7}$$

where $\boldsymbol{\epsilon} \sim \mathcal{N}(0, \boldsymbol{I})$. This process can be repeated for a set number of steps, gradually reducing the noise level until a final solution is obtained.

## 3.2 ONE-STEP DIFFUSION SOLVER WITH CONSISTENCY DISTILLATION

To achieve one-shot inference, a consistency model $f_{\theta'}$, is distilled from the pre-trained and frozen teacher model $f_\theta$. The distillation process trains the student model $f_{\theta'}$ to map any point $(\boldsymbol{z}_t, t)$ on a trajectory to its endpoint $\boldsymbol{z}_0$. This is achieved by enforcing consistency between the outputs of adjacent points on a single trajectory, following the methodology in your implementation. Specifically, for each training step:

We sample a discrete time step $n \in \{1, \ldots, T-1\}$ and define two adjacent noisy states on the same trajectory: $\boldsymbol{z}_t$ corresponding to the noisier time $t = n + 1$, and $\boldsymbol{z}_{t_{\text{next}}}$ corresponding to the less noisy time $t_{\text{next}} = n$.

The frozen teacher model $f_\theta$ provides a target prediction by denoising the less noisy state $\boldsymbol{z}_{t_{\text{next}}}$:

$$\hat{\boldsymbol{z}}_0^{\text{teacher}} = f_\theta(\mathcal{G}, \boldsymbol{z}_{t_{\text{next}}}, t_{\text{next}}). \tag{8}$$

The student model (the consistency model) $f_{\theta'}$ makes its own prediction from the noisier state $\boldsymbol{z}_t$:

$$\hat{\boldsymbol{z}}_0^{\text{student}} = f_{\theta'}(\mathcal{G}, \boldsymbol{z}_t, t). \tag{9}$$

The parameters $\theta'$ of the student model are updated by minimizing the Mean Squared Error between its prediction and the teacher's target, ensuring consistency along the trajectory:

$$\mathcal{L}_{\text{distill}} = \mathbb{E}_{\boldsymbol{z}_0, n, \boldsymbol{\epsilon}}\left[\alpha \underbrace{\|f_{\theta', x}(\mathcal{G}, \boldsymbol{z}_t, t) - f_{\theta, x}(\mathcal{G}, \boldsymbol{z}_{t_{\text{next}}}, t_{\text{next}})\|^2}_{\text{Primal Variable Loss}}\right.$$
$$\left. + (1 - \alpha) \underbrace{\|f_{\theta', y}(\mathcal{G}, \boldsymbol{z}_t, t) - f_{\theta, y}(\mathcal{G}, \boldsymbol{z}_{t_{\text{next}}}, t_{\text{next}})\|^2}_{\text{Dual Variable Loss}}\right]. \tag{10}$$

This distillation-based consistency training enables single-step generation from any noise level. The core solver of the consistency model is implemented using a Graph Neural Network (GNN), whose structure will be detailed in the next section.

In summary, by reformulating optimization as a generative task and leveraging consistency training to accelerate inference, this framework offers a novel, efficient, and theoretically grounded solution for solving QP problems.

### 3.3 KKT GUIDANCE METRIC

The goal of guidance techniques in diffusion models is to enhance controllability, steering the generation process not only to match the data distribution but also to satisfy specific attributes or constraints. Early methods such as Classifier Guidance Dhariwal & Nichol (2021) employed an auxiliary classifier to provide gradients of log-likelihood with respect to a desired class, significantly improving both quality and controllability, though at the cost of training an additional model and risks of overfitting. The subsequent Classifier-Free Guidance (CFG) Ho & Salimans (2022) eliminated this dependency by jointly training a single diffusion model with conditional and unconditional objectives, and extrapolating between the two at inference time. CFG has since become the mainstream approach for controllable image, speech, and text generation.

Building on this, researchers introduced more general Energy-Based Guidance Liu et al. (2022), which no longer relies on an explicit classifier but instead uses a differentiable energy function to evaluate whether generated samples satisfy desired properties. The gradient of this energy serves as the steering signal. This paradigm has inspired variants such as Reward Guidance or Reinforcement Learning-Guided Diffusion, which optimize generations against a learned reward model or human preference, as well as Contrastive Guidance, which leverages contrastive learning signals. Together, these developments reflect the broader trend of diffusion models evolving from unconditional generation towards goal-driven generation.

In this work, we present **KKT Guidance**, which can be seen as a domain-specific instance of energy-based guidance Guo et al. (2024) tailored to constrained optimization. Importantly, KKT Guidance is applied only during inference of the **DDPM teacher model**, not during training of the student consistency model. At each sampling step, the teacher model predicts a candidate solution $\hat{z}_0 = [\hat{x}_0, \hat{y}_0]^T$, where $\hat{x}_0$ are the primal variables and $\hat{y}_0$ are the dual variables. A differentiable energy function is then used to measure the violation of the Karush-Kuhn-Tucker (KKT) conditions:

$$
\begin{aligned}
E(z) = {} & \lambda_{\text{peq}} \, \|A\hat{x}_0 - b\|^2 + \lambda_{\text{pineq}} \, \| \max(0, -\hat{x}_0)\|^2 \\
& + \lambda_{\text{dineq}} \, \| \max(0, -(Q\hat{x}_0 + c + A^T \hat{y}_0))\|^2 \\
& + \lambda_{\text{cs}} \, \|\hat{x}_0 \odot (Q\hat{x}_0 + c + A^T \hat{y}_0)\|^2 .
\end{aligned}
\tag{11}
$$

This formulation encompasses equality constraint residuals, inequality constraint residuals, dual feasibility, and complementary slackness. The energy reaches zero if and only if the candidate solution fully satisfies the KKT conditions.

To enforce feasibility, the gradient of the energy with respect to the solution is computed, and the prediction is corrected as:

$$
\tilde{z}_0 = \hat{z}_0 - s \cdot \nabla_z E(z)\big|_{z=\hat{z}_0},
$$

where $s$ is the guidance scale hyperparameter. The corrected solution $\tilde{z}_0$ is then passed to the ODE solver for the next time step. By iteratively performing this "predict–evaluate–correct" cycle, the generation trajectory is guided into the feasible region while approaching optimality.

The benefits of this approach are clear: the generation process is no longer limited to mimicking the training data distribution, but becomes a solution process with a well-defined objective. Specifically, KKT Guidance does not rely solely on external models to evaluate the quality of generated outputs. Instead, it directly uses the mathematical conditions for optimality (the KKT conditions) as the guiding principle, ensuring the diffusion model naturally adheres to these constraints during generation.

### 3.4 GNN FOR QUADRATIC PROGRAMMING

The QP instance can be represented as a bipartite graph, which is brought up by Chen et al. (2024). And this paper provides a detailed theoretical analysis of the representational capabilities of GNNs, which offers significant inspiration for the content of our work. The GNN operates on a bipartite graph consisting of variable and constraint nodes, and employs a multi-layer message-passing scheme where information alternates between the two sets of nodes. To adapt this GNN to the diffusion framework, we introduce some modifications. The details are in Appendix A.

### 3.5 PARALLEL POST-PROCESSING COMPUTATIONS

Applying deep learning models to quadratic programming problems is an effective way to quickly obtain approximate solutions. However, the outputs of such data-driven methods often fail to strictly satisfy the constraints. To address this, we propose a post-processing framework that leverages the KKT conditions and the structure of traditional algorithms to correct these initial guesses. The objective is to generate a new point $(x_{\text{corrected}}, y_{\text{corrected}})$ with improved consistency with the problem's optimality criteria. Within this framework, we present two classes of approaches.

**ADMM Refinement.** The first category comprises primal-dual algorithms from traditional optimization (e.g., ADMM, PDHG, ALM) in Appendix D, designed to fully leverage the primal and dual variable information output by the diffusion model. Taking our proposed ADMM-based approach as an example, the complete predicted solution pair $(x_{\text{init}}, y_{\text{init}})$ generated by the diffusion model is directly used as the initial starting point for the algorithm. To ensure numerical stability and computational correctness, each x-update step within the ADMM iteration is performed by solving the corresponding KKT linear system, which concurrently yields the optimal primal and dual variables for the subproblem. This design facilitates a seamless transition from the generative model to the iterative solver, effectively leveraging the prior knowledge acquired by the model in the primal-dual space to achieve accelerated convergence and high-fidelity feasible solutions.

**Iterable KKT Corrector.** The second class consists of correction strategies that directly exploit the KKT conditions. This method assumes that the primal solution $x_{\text{init}}$ is a reliable estimate, and seeks a dual solution $y$ that is mathematically consistent with it. The approach is motivated by iterative application of the KKT stationarity condition $Qx + c + A^T y - z = 0$ together with the dual feasibility condition $z \geq 0$. The detailed derivation of the two methods is provided in Appendix C.

In summary, these two approaches—refinement through traditional solvers and KKT-based correction provide complementary pathways for bridging the gap between fast, data-driven approximations and the rigorous optimality requirements of classical optimization.

## 4 EXPRIMENTS

All experiments were conducted on a single NVIDIA RTX 4090 GPU. The code will be made publicly available to ensure reproducibility. For detailed guarantees, please refer to section 5.

### 4.1 DATASETS

The dataset utilized in this study was specifically designed for quadratic programming (QP) problems, with its generation Gondzio (1997) process following a rigorous procedure to ensure data quality and verifiability. The test set was specifically designed to include both dense and sparse Q matrices. Both sparse and dense Q have their own application scenarios Boykov & Veksler (2006),Platt (1998). The dataset contains 1,280 instances for training and is evaluated on a separate test set of 128 instances. Each instance in the dataset adopts the standard form in Eq. 1. The test cases in the experiment were processed one by one without multiple instance computation (i.e. batchsize is set to 1 in inference process).

### 4.2 EVALUATION AND BASELINES

We compare the objective function value obtained by different solvers, denoted as $f(\boldsymbol{x})$, with the ground-truth optimal value, $f(\boldsymbol{x}_{\text{true}})$. The objective function $f(\boldsymbol{x})$ for the quadratic program is defined as: $f(\boldsymbol{x}) = \frac{1}{2}\boldsymbol{x}^\top Q \boldsymbol{x} + \boldsymbol{c}^\top \boldsymbol{x}$.

Based on this, we evaluate the accuracy using the following metrics:

**Relative Cost Gap (Gap)**: To account for the scale of the objective values, we compute the average of instance-wise relative errors. Each error is normalized by the magnitude of the true objective value, with a small constant $\epsilon$ added for numerical stability:

$$\text{Gap}_{\text{rel}} = \mathbb{E}\left[\frac{|f(\boldsymbol{x}) - f(\boldsymbol{x}_{\text{true}})|}{|f(\boldsymbol{x}_{\text{true}})| + \epsilon}\right]. \tag{12}$$

Table 1: Results on sparse Q instances with 100 variables and 80 constraints.

| Method | Obj | MAE | Gap (%) | Time (s) | PrimalRes | DualRes | Solved |
|---|---|---|---|---|---|---|---|
| Real (Optimal) | -17.40 | - | - | - | - | - | - |
| Gurobi-TimeLimit | -17.40 | - | - | 1.30 | - | - | all |
| ADMM | -17.56 | 1.00 | 5.5 | 7.80 | 1.26 | 0.57 | all |
| PDHG | -17.16 | 1.78 | 5.4 | 2.79 | 1.75 | 2.30 | all |
| GNN | -17.75 | 1.6 | 22.9 | 0.5 | 0.7 | 4.1 | all |
| PDDQP | -17.01 | 1.00 | 8.0 | 1.57 | 2.00 | 1.10 | all |
| PDDQP-R | -17.41 | 0.02 | 0.2 | 2.42 | 0.00 | 0.06 | all |

Table 2: Results on sparse Q instances with 500 variables and 100 constraints.

| Method | Obj | MAE | Gap (%) | Time (s) | PrimalRes | DualRes | Solved |
|---|---|---|---|---|---|---|---|
| Real (Optimal) | -32.22 | - | - | - | - | - | - |
| Gurobi -TimeLimit | -32.22 | - | - | 113 | - | - | 65/113 |
| ADMM | -34.5 | 2.32 | 7.0 | 10.21 | 1.0 | 0.3 | all |
| PDHG | -30.5 | 1.78 | 5.5 | 4.26 | 1.7 | 3.0 | all |
| GNN | -2.3 | 30.0 | 99 | 1.92 | 2.9 | 10.2 | all |
| PDDQP | -32.29 | 0.79 | 3.0 | 1.60 | 2.0 | 7.3 | all |
| PDDQP-R | -32.25 | 0.066 | 0.2 | 3.74 | 0 | 0.14 | all |

**Primal Residual** ($r_{\mathbf{primal}}$): The feasibility of a solution is measured by the degree of violation of the equality constraints $A\boldsymbol{x} = \boldsymbol{b}$, quantified using the L2-norm:

$$r_{\mathrm{primal}} = \|A\boldsymbol{x} - \boldsymbol{b}\|_2. \tag{13}$$

**Dual Residual** ($r_{\mathbf{dual}}$): This metric quantifies the violation of the dual feasibility conditions, serving as a measure of optimality. Its significance lies in ensuring that no feasible direction at the current solution can further reduce the objective function, providing a necessary gradient-based condition for optimality:

$$r_{\mathrm{dual}} = \left\|\max\left(0, -\left(Q\boldsymbol{x} + \boldsymbol{c} + A^T\boldsymbol{y}\right)\right)\right\|_2. \tag{14}$$

**Computational Efficiency**: The total wall-clock time required to solve all test instances is measured to evaluate computational efficiency. All problems are processed sequentially.

We compare the proposed method with various techniques, including the industrial-grade optimizer Gurobi, conventional optimization algorithms such as ADMM Boyd et al. (2011) and PDHG He et al. (2014)and PDHG, and a supervised learning approach based on Graph Neural Networks (GNN). Besides, our proposed PDDQP-R (refinement) is obtained by integrating one-step sampling with a few iterations of the KKT corrector, followed by a fixed number (10) of ADMM iterations.

## 4.3 EXPERIMENTAL RESULTS

Our PDDQP model is capable of rapidly generating high-quality solutions. Even without subsequent refinement steps, the model maintains strong performance. When combined with post-optimization procedures, solution quality can be further enhanced with just a few iterations. Particularly on relatively high-dimensional data, PDDQP-R can produce solutions with near-zero constraint violations (on the order of $10^{-5}$), indicating that the solutions found by the model lie almost entirely within the feasible region defined by the constraints $A\mathbf{x} = \mathbf{b}$.

Graph Neural Networks (GNNs) underperform in high-dimensional scenarios and even fail to converge entirely in cases involving dense graphs with, for example, 200 variables and 80 constraints. Moreover, solutions output directly by GNNs may exhibit significant constraint violations, sometimes even finding solutions with a lower objective function value but at the cost of feasibility. While GNNs show considerably better performance on some simpler datasets (a detailed discussion is provided in Appendix E), their performance on such datasets still falls far short of Gurobi's.

Table 3: Results on dense Q instances with 50 variables and 40 constraints.

| Method | Obj | MAE | Gap (%) | Time (s) | PrimalRes | DualRes | Solved |
|---|---|---|---|---|---|---|---|
| Real (Optimal) | -31.73 | - | - | - | - | - | all |
| Gurobi-TimeLimit | -31.73 | - | - | 19.0 | - | - | all |
| ADMM | -31.76 | 0.30 | 0.9 | 2.10 | 0.25 | 0.10 | all |
| PDHG | -31.54 | 0.60 | 1.8 | 2.60 | 0.95 | 1.56 | all |
| GNN | -28.9 | 3.05 | 13.4 | 0.4 | 0.58 | 2.0 | all |
| PDDQP | -31.57 | 0.30 | 0.9 | 1.00 | 0.60 | 6.10 | all |
| PDDQP-R | -31.72 | 0.05 | 0.2 | 1.62 | 0.12 | 1.26 | all |

Table 4: Results on dense Q instances with 200 variables and 80 constraints.

| Method | Obj | MAE | Gap (%) | Time (s) | PrimalRes | DualRes | Solved |
|---|---|---|---|---|---|---|---|
| Real (Optimal) | -147.40 | - | - | - | - | - | - |
| Gurobi-TimeLimit | -147.40 | - | - | 65 | - | - | 75/128 |
| ADMM | -146.18 | 2.16 | 7.0 | 10.19 | 1.36 | 0.48 | all |
| PDHG | -146.18 | 1.78 | 5.5 | 4.26 | 2.3 | 4.7 | all |
| PDDQP | -147.62 | 0.93 | 0.6 | 1.60 | 1.88 | 6.4 | all |
| PDDQP-R | -147.40 | 0.05 | 0.03 | 2.96 | 0 | 0.13 | all |

Gurobi undoubtedly produces the most reliable solutions with virtually zero constraint violation and achieves remarkable speed on low-dimensional problems. However, its performance degrades significantly in high-dimensional settings, which is precisely where PDDQP demonstrates its advantages.TimeLimit indicates that Gurobi failed to find all optimal solution within a relatively long time (1–2 minutes).

Traditional algorithms, such as ADMM and PDHG, are highly dependent on the selection of initial parameters and require a relatively large number of iterations to converge to a valid solution, thus resulting in slightly inferior performance. Nevertheless, these traditional algorithms are straightforward, interpretable, and their potential remains worthy of further exploration.

## 5 CONCLUSION AND FUTURE WORK

In previous work, dual variables are not directly predicted but are instead utilized to construct Lagrangian functions for reformulating the optimization objective. Through empirical validation, this study confirms that it is entirely feasible to generate high-quality dual variables without compromising the quality of the primal variables, thereby providing multiple options for subsequent refinement. We argue that the methodology presented in this paper can be extended as a novel training paradigm to other bipartite graph tasks, and even to graph problems involving multiple node types. Our PDDQP method was originally designed to fully leverage the information from both types of nodes in such representations. The primary rationale for employing diffusion models lies in their ability to progressively refine high-quality solutions from noise, which helps circumvent the limitations of one-shot predictions inherent in GNNs and demonstrates greater potential in terms of solution robustness and accuracy. Furthermore, our method aims to harness the advantages of traditional algorithms, particularly their high precision and convergence guarantees.

We believe that future improvements to this work should focus on two key aspects. First, the highest priority should be given to modifications of the Graph Neural Network architecture. For instance, incorporating self-attention mechanisms could enhance model performance Veličković et al. (2017). Second, regarding post-optimization methods, while the currently employed ADMM algorithm is effective and straightforward, integrating the post-optimization process into the network architecture for end-to-end training may yield superior results.

## ETHICS STATEMENT

This paper aims to advance the state of the art in machine learning and artificial intelligence for electronic design automation (AI4EDA). While the research may entail various societal implications, we do not identify any that warrant specific emphasis in this paper.

## REPRODUCIBILITY STATEMENT

All experimental results in the paper are reproducible, and the implementation code for reproducing experimental results will be fully open sourced on Github after the paper is accepted.

## LLM USAGE STATEMENT

The contribution of LLM in the work proposed in this article is limited to: 1. polishing given written statements; 2. Given written sentence syntax review. We declare that no experimental data was generated/modified by LLM.

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

# A GNN

To adapt this GNN to the diffusion framework, we introduce two key modifications. First, to make the model aware of the current noisy state, the noisy primal and dual variables $(\boldsymbol{x}_t, \boldsymbol{y}_t)$ are concatenated with the static node features at the input layer, augmenting the representation of each node with the current solution estimate. Second, to make the model time-aware, the diffusion time step $t$ is mapped into a high-dimensional vector $e_t$ via a dedicated embedding module. This embedding is then added to the hidden states of both variable and constraint nodes at every layer, allowing the network to modulate its denoising strength according to the noise level.In the original GNN paper, upper and lower bound information was incorporated for each variable node. However, in our task, due to the inclusion of dual variables and time steps (primarily the dual variables), this means the bounds can only correspond to a subset of the input information. Therefore, we have removed them.

Formally, let $h_v^{(l)}$ and $h_c^{(l)}$ denote the features of variable and constraint nodes at layer $l$. The update from variables to constraints is given by

$$h_c^{(l+1)} = \text{Update}_c\left(h_c^{(l)}, \sum_{v \in \mathcal{N}(c)} A_{cv}\, g_v(h_v^{(l)}), e_t\right), \tag{15}$$

where $\text{Update}_c$ applies residual connections, normalization, and time embedding injection. Conversely, the update for a variable node $v$ incorporates both constraint messages and quadratic couplings with neighboring variables:

$$h_v^{(l+1)} = \text{Update}_v\left(h_v^{(l)}, \sum_{c \in \mathcal{N}(v)} A_{cv}\, g_c(h_c^{(l+1)}), \sum_{v' \in \mathcal{N}(v)} Q_{vv'}\, g_v(h_{v'}^{(l)}), e_t\right). \tag{16}$$

After the final message-passing layer, the network prepares the node embeddings for prediction. A global context vector is first obtained by aggregating features across all nodes, which is then concatenated with each node's local representation. Finally, two parallel MLP output heads are applied: one predict the primal variables $\boldsymbol{x}$ from variable node embeddings, while the other predict the dual variables $\boldsymbol{y}$ from constraint node embeddings.

# B DATA GENERATION

Initially, the core parameters of the problems are generated randomly. The matrix $Q$ is constructed as a sparse, positive-definite, and symmetric matrix by invoking the make_sparse_spd_matrix function from the scikit-learn library, with its sparsity precisely controlled by a parameter. The matrix $A$ is set as a random sparse matrix, and the non-zero ratio of A is 10%. To guarantee that a feasible solution exists for every instance, we first generate a random feasible solution vector $x_{\text{feas}}$ with non-negative elements, and then determine the equality constraint vector $b$ by computing $b = Ax_{\text{feas}}$. Finally, all successfully solved problem instances with their exact labels and parameters are saved and categorized, forming the complete dataset for subsequent research.

---

**Algorithm 1 Data generation method**

---

1: **Input:** Problem dimensions $m, n$; non-zero count $nnz$.
2: **Output:** QP instance $(Q, A, b, c)$ and its optimal solution $x$.
3: **Generate**:
  Sparse matrix $A \in \mathbb{R}^{m \times n}$ with $nnz$ non-zero entries.
  Symmetric positive definite matrix $Q \in \mathbb{R}^{n \times n}$.
  Primal solution $\hat{x} \in \mathbb{R}_+^n$.
  Dual variables $(\hat{y}, \hat{s}) \in \mathbb{R}^m \times \mathbb{R}_+^n$ such that KKT conditions hold.
4: $b \leftarrow A\hat{x}$
5: $c \leftarrow A^\top \hat{y} + \hat{s} - Q\hat{x}$
6: **Validate:** Solve the QP instance $(Q, A, b, c)$ to ensure $(\hat{x}, \hat{y}, \hat{s})$ are optimal.

---

---

**Algorithm 2** Iterable KKT Consistency Corrector

---

1: **Input:** Initial solution $(x_{\text{init}}, y_{\text{init}})$, problem data $(Q, c, A)$, number of iterations $K$.
2: **Output:** Corrected dual solution $y_{\text{corrected}}$.
3: **Pre-computation:**
4:     Compute the static term: $S \leftarrow Qx_{\text{init}} + c$.
5:     Compute the pseudo-inverse of the transposed constraint matrix: $(A^T)^\dagger$.
6: **Initialize:** $y_0 \leftarrow y_{\text{init}}$.
7: **for** $k = 0, 1, \ldots, K - 1$ **do**
8:     Compute implied dual slack: $z_k \leftarrow S + A^T y_k$.
9:     Project to enforce dual feasibility: $z_{k+1} \leftarrow \max(0, z_k)$.
10:     Update dual variable using pre-computed inverse: $y_{k+1} \leftarrow (A^T)^\dagger (z_{k+1} - S)$.
11: **end for**
12: **Return** $y_{\text{corrected}} \leftarrow y_K$.

---

## C  KKT-BASED REFINEMENT

## D  CLASSIC METHOD REFINEMENT

**Alternating Direction Method of Multipliers (ADMM).** This algorithm addresses the QP problem in Equation 1 by reformulating it with a splitting variable $z$ and a consensus constraint $x = z$. This allows for decoupling the non-negativity constraint, leading to the equivalent formulation:

$$\min_{x,z} \quad \tfrac{1}{2}x^T Q x + c^T x + \mathcal{I}_{\geq 0}(z)$$
$$\text{subject to} \quad Ax = b, \quad x - z = 0, \tag{17}$$

where $\mathcal{I}_{\geq 0}(z)$ is the indicator function for the non-negative cone. The augmented Lagrangian for the consensus constraint is:

$$L_\rho(x, z, u) = \tfrac{1}{2}x^T Q x + c^T x + u^T (x - z) + \tfrac{\rho}{2}\|x - z\|_2^2,$$

where $u$ is the dual variable for the consensus constraint. Our hybrid approach combines the strengths of data-driven initialization and the rigorous convergence properties of classical operator splitting methods.

**Primal-Dual Hybrid Gradient (PDHG).** This algorithm operates on the primal-dual optimality conditions derived from the problem's Lagrangian. The Lagrangian for this problem (ignoring the non-negativity constraint, which will be handled by a proximal step) is:

$$L(x, y) = \tfrac{1}{2}x^T Q x + c^T x + y^T (Ax - b) \tag{18}$$

where $y$ is the dual variable associated with the equality constraint $Ax = b$. PDHG is an iterative method that alternates between a gradient ascent step on the dual variable $y$ and a proximal gradient descent step on the primal variable $x$. The implementation in the code also includes an extrapolation step on the primal variable to accelerate convergence. In the $(k + 1)$-th iteration:

**Augmented Lagrangian Method (ALM).** This algorithm iteratively solves a series of unconstrained (or simply constrained) subproblems to find a solution to the original constrained problem. The QP optimization problem being solved has the following standard form in Equation 1.

The core of the ALM is the augmented Lagrangian function, which incorporates the equality constraints into the objective function using both a Lagrange multiplier term and a quadratic penalty term. The function is defined as:

$$L_\rho(x, y) = \tfrac{1}{2}x^T Q x + c^T x + y^T (Ax - b) + \tfrac{\rho}{2}\|Ax - b\|_2^2 \tag{19}$$

where $y$ is the vector of Lagrange multipliers (the dual variables) associated with the constraint $Ax = b$, and $\rho > 0$ is the penalty parameter. The ALM algorithm alternates between minimizing $L_\rho$ with respect to the primal variable $x$, and then updating the dual variable $y$ and the penalty parameter $\rho$. In the $(k + 1)$-th iteration:

---

**Algorithm 3** ADMM for QP Refinement with Dual Variable Initialization

---
][t]

1: **Input:** Initial primal solution $x_{\text{init}}$, initial dual solution $y_{\text{init}}$, problem data $(Q, c, A, b)$, penalty parameter $\rho > 0$, tolerance $\epsilon > 0$.
2: **Output:** Refined primal solution $x$, refined dual solution $y$.
3: **Initialize:** $k \leftarrow 0$, $x_0 \leftarrow x_{\text{init}}$, $y_0 \leftarrow y_{\text{init}}$, $z_0 \leftarrow \max(0, x_0)$, $u_0 \leftarrow \mathbf{0}$.
4:
5: **Pre-computation:** Construct the constant KKT matrix:

$$K \leftarrow \begin{bmatrix} Q + \rho I & A^T \\ A & 0 \end{bmatrix}$$

6: **while** not converged **do**
7:   **x, y-update:** Solve the linear KKT system for $x_{k+1}$ and $y_{k+1}$ simultaneously:

$$K \begin{bmatrix} x_{k+1} \\ y_{k+1} \end{bmatrix} = \begin{bmatrix} \rho z_k - u_k - c \\ b \end{bmatrix}$$

8:   **z-update:** $z_{k+1} \leftarrow \max\left(0, x_{k+1} + \frac{1}{\rho} u_k\right)$ {Projection step for non-negativity}
9:   **u-update:** $u_{k+1} \leftarrow u_k + \rho(x_{k+1} - z_{k+1})$ {Dual update for the splitting constraint}
10:   **Check convergence:**
11:   **if** $k > 0$ **and** $\|x_{k+1} - x_k\|_2 \le \epsilon$ **and** $\|x_{k+1} - z_{k+1}\|_2 \le \epsilon$ **then**
12:     **break**
13:   **end if**
14:   $k \leftarrow k + 1$
15: **end while**
16: **return** $x_{k+1}, y_{k+1}$

---

---

**Algorithm 4** PDHG for QP

---

1: **Input**: Problem data $(Q, c, A, b)$, step sizes $\tau, \sigma > 0$, tolerances $\epsilon_{\text{primal}}, \epsilon_{\text{change}} > 0$.
2: **Output**: Solution $x$.
3: **Initialize**: $k \leftarrow 0$, compute initial $x_0$ (e.g., via pseudoinverse $A^\dagger b$), $y_0 \leftarrow \mathbf{0}$, $\bar{x}_0 \leftarrow x_0$.
4: **while** not converged **do**
5:   Dual update (gradient ascent):
6:

$$y_{k+1} \leftarrow y_k + \sigma(A\bar{x}_k - b)$$

7:   Primal update (proximal gradient descent): Solve the subproblem for $x_{k+1}$:
8:

$$x_{k+1} \leftarrow \arg\min_{x \ge 0} \left\{ \tfrac{1}{2} x^T Q x + c^T x + y_{k+1}^T A x + \frac{1}{2\tau} \|x - x_k\|_2^2 \right\}$$

9:   Extrapolation step:
10:

$$\bar{x}_{k+1} \leftarrow x_{k+1} + (x_{k+1} - x_k)$$

11:   Check convergence:
12:   **if** $k > 10$ and $\|Ax_{k+1} - b\|_2 \le \epsilon_{\text{primal}}$ and $\|x_{k+1} - x_k\|_2 \le \epsilon_{change}$ **then**
13:     **break**
14:   **end if**
15:   $k \leftarrow k + 1$
16: **end while**

---

---

**Algorithm 5** ALM for QP

---

1: **Input:** Problem data $(Q, c, A, b)$, initial penalty $\rho_0 > 0$, tolerances $\epsilon_{\text{primal}}, \epsilon_{\text{change}} > 0$.
2: **Output:** Solution $x$.
3: **Initialize:** $k \leftarrow 0$, $x_0 \leftarrow \mathbf{0}$, $y_0 \leftarrow \mathbf{0}$, $\rho_0$.
4:
5: **while** not converged **do**
6:     x-update (Primal Minimization): Solve the subproblem for $x_{k+1}$. The non-negativity constraint $x \geq 0$ is enforced after this minimization step (e.g., by projection).

$$x_{k+1} \leftarrow \text{projection}_{\geq 0} \left( \arg \min_x L_{\rho_k}(x, y_k) \right)$$

7:     y-update (Dual Update):
$$y_{k+1} \leftarrow y_k + \rho_k (Ax_{k+1} - b)$$

8:     $\rho$-update (Penalty Parameter Update):
9:     **if** the constraint violation $\|Ax_{k+1} - b\|_2$ is not improving sufficiently **then**
10:         $\rho_{k+1} \leftarrow \rho_k \cdot \text{factor}$
11:     **else**
12:         $\rho_{k+1} \leftarrow \rho_k$
13:     **end if**
14:     **Check convergence:**
15:     **if** $k > 0$ **and** $\|Ax_{k+1} - b\|_2 \leq \epsilon_{\text{primal}}$ **and** $\|x_{k+1} - x_k\|_2 \leq \epsilon_{\text{change}}$ **then**
16:         break
17:     **end if**
18:     $k \leftarrow k + 1$
19: **end while**

---

# E LOW-DIMENSIONAL TESTING

According to the paper Chen et al. (2024), this dataset demonstrates relatively strong performance for GNNs. Since there are certain discrepancies between our data and those in the paper, to ensure rigor, our experimental results are largely consistent with the performance reported in the original study on this type of dataset. However, testing on such a dataset fails to fully showcase the model's powerful capabilities. The main reason is that when the number of constraints is small, without strictly controlling constraint violations, the process essentially resembles sampling in an unconstrained space. Even if treated as an unconstrained problem, the obtained solution would not deviate significantly. Moreover, for such problems, industrial-grade optimizer like Gurobi can easily handle them and deliver valid solutions within a very short time.

Table 5: Results on sparse Q instances (50 variables, 10 constraints)

| Method | Obj | MAE | Gap (%) | Time (s) | PrimalRes | PrimalRes | Solved |
|--------|-----|-----|---------|----------|-----------|-----------|--------|
| Real (Optimal) | -1.61 | - | - | - | - | - | - |
| Gurobi | -1.61 | - | - | 0.52 | - | - | all |
| PDDQP-R | -1.61 | 0.01 | 1.5 | 1.56 | 0.02 | 0.08 | all |
| PDDQP | -1.55 | 0.10 | 23.0 | 1.23 | 0.05 | 0.32 | all |
| ADMM | -1.63 | 0.06 | 7.0 | 2.00 | 0.03 | 0.11 | all |
| PDHG | -1.59 | 0.08 | 5.0 | 2.50 | 0.03 | 0.12 | all |
| GNN | -1.59 | 0.09 | 4.8 | 0.38 | 0.04 | 0.19 | all |

# F  OTHER REFINEMENT METHOD TEST

Selecting an effective post-refinement method is crucial for PDDQP. Although ALM, as a classical algorithm, possesses theoretical convergence guarantees, it suffers from practical limitations such as high sensitivity to penalty parameter selection and numerical instability. These issues often prevent convergence to the optimal solution after multiple iterations. However, with the improved initial point provided by PDDQP, ALM demonstrates significantly enhanced performance.

Table 6: Results on sparse Q instances (500 variables, 100 constraints, 5000 nnz)

| Method | Obj | MAE | Gap (%) | Time (s) | PrimalRes | DualRes | Solved |
|---|---|---|---|---|---|---|---|
| Real (Optimal) | -32.22 | - | - | - | - | - | - |
| Gurobi | - | - | - | * | - | - | 65/113 |
| PDDQP-R | -31.68 | 0.54 | 2.0 | 1.81 | 0.3 | - 0.14 | all |
| PDDQP | -32.29 | 0.79 | 3.0 | 1.60 | 2.0 | 7.3 | all |
| ALM | -27.27 | 5.0 | 15.2 | 34.3 | 0.028 | 0.14 | all |

