# OpenReview forum: "Primary-Dual Diffusion Solver for Quadratic Programming Problems"
_ICLR.cc/2026/Conference — ICLR 2026 Conference Withdrawn Submission_

### Official Review · Reviewer_ZMb3 · 2025-10-31

**Soundness:** 2
**Presentation:** 2
**Contribution:** 2
**Rating:** 2
**Confidence:** 3

**Summary:**

The paper proposes a PDD framework for QP, generating primal and dual variables with a DDPM teacher and distilling a one-step student via consistency training while enforcing feasibility through KKT-guided sampling. The final solution is then obtained by several iterations of classical refinements, such as ADMM or PDHG.

**Strengths:**

1. The design of the PDD structure aligns the learned initializer with downstream optimality conditions and provides a principled handle to reduce projection burdens during refinement.
2. The one-step student offers a latency reduction while preserving numerical priors learned by the teacher. This may help to decrease the number of ADMM/PDHG steps needed in the final step.

**Weaknesses:**

1. The empirical evidence is confined to synthetic QP generators and moderate scales, so the claimed “growing advantage at higher dimensions” is insufficiently substantiated for large-scale QP problems.
2. The method strongly relies on post-refinement, and the statements about “obtaining the optimal solution” are not backed by standalone guarantees for the learned component nor by ablations that separate the contributions of teacher sampling, KKT guidance, and classical iterations. In general, there is no condition where the diffusion initializer alone reliably reaches near-optimality, and no clean ablations separating the learned part from refiners.
3. No convergence, feasibility, or stability analysis is provided. KKT guidance in this structure seems a heuristic sampling of energy rather than a guarantee.
4. The authors do not report training cost, memory footprint, inference latency, and required refinement steps. They also do not provide end-to-end elapsed time comparisons at matched accuracy with warm-started solvers. Including training and maintenance may erode any claimed speedups.
5. There is no systematic breakdown of gains from primal–dual joint generation, KKT guidance, consistency distillation, and the GNN backbone. Improvements could largely stem from the classical refinements, leaving the diffusion stage’s true contribution unclear.

**Questions:**

N/A

---

### Official Review · Reviewer_e9n5 · 2025-10-31

**Soundness:** 1
**Presentation:** 1
**Contribution:** 2
**Rating:** 2
**Confidence:** 3

**Summary:**

The paper proposes a diffusion-based neural QP solver that jointly predicts primal and dual variables, distills to a one-step model, and applies KKT-based refinement. Claims include both high accuracy and competitive runtime vs traditional solvers.

**Strengths:**

1. Joint primal–dual learning. Conceptually aligned with KKT structure in QP.
2. Attempts to combine learning with classical optimization.
3. Use of diffusion-based generative modeling in a continuous constrained optimization setting may be novel.

**Weaknesses:**

Motivation vs. chosen architecture mismatch. The paper frames speed as a core motivation, yet diffusion inference is inherently multi-step and computationally heavy. Distillation is proposed but lacks rigorous evidence of preserving solution quality under reduced steps.
Distillation motivation and roles unclear. Paper does not articulate why a teacher diffusion model is needed, nor provide ablations showing student-only training fails. It remains unclear what optimization knowledge is actually distilled.

**Questions:**

Experimental design insufficient to justify claims. Experiments are small-scale toy QPs (≤500 variables). Wall-clock timing comparisons are not controlled or fair because performance gaps fall within configuration variability.

Problem formulation and guidance contain non-differentiable terms. The “KKT Guidance” uses max(·) in (11), making gradients undefined at zero. This contradicts diffusion’s requirement for smooth gradients. No workaround or theoretical analysis is provided.

Notation errors and undefined symbols. Multiple symbols appear without definition (e.g., PDDDP, \theta). Several repeated sentences and formatting errors throughout the manuscript indicate immaturity of the submission.

In Appendix D, the PDHG method of (18) and the ALM method of (19) cannot handle inequality constraints where x>=0 in the proplem (1)

---

### Official Review · Reviewer_vckv · 2025-11-02

**Soundness:** 2
**Presentation:** 2
**Contribution:** 2
**Rating:** 4
**Confidence:** 3

**Summary:**

The paper proposes a Primary–Dual Diffusion (PDD) framework for convex quadratic programs (QP). The authors claim the approach attains near-optimal solutions with only a few correction iterations and scales favorably versus commercial solvers at higher dimensions. Experiments are reported on synthetic QP instances with metrics including relative objective gap, primal/dual residuals, and wall-clock time.

**Strengths:**

Predicting solutions aligns with KKT structure and gives good warm starts for classical methods. The architectural choice (bipartite GNN over variable/constraint nodes) is well-motivated.

A clear distillation recipe for fast inference; the paper argues why other one-step generative variants underperform for numerically sensitive QP.

**Weaknesses:**

The core ingredients—diffusion for solution generation, a GNN over primal/dual factor graphs, and a KKT-residual (feasibility) loss—are each well-established in adjacent literature. As presented, the paper’s contribution seems more like a careful composition/tuning of known pieces rather than a clearly new principle. The manuscript would benefit from a sharper positioning of what is genuinely novel (algorithmically or theoretically) versus what is inherited.

**Questions:**

How does the model behave under distribution shift in (Q,A,b,c) especially sparsity patterns and conditioning of 𝑄

---

### Official Review · Reviewer_hRiP · 2025-11-11

**Soundness:** 1
**Presentation:** 2
**Contribution:** 2
**Rating:** 2
**Confidence:** 5

**Summary:**

This paper presents a diffusion-based model for solving quadratic programming (QP) problems that jointly learns both primal and dual variables. The core method involves training a one-step solver through consistency distillation from a multi-step teacher model, with KKT-based energy guidance to encourage feasibility during sampling. Experimental results show promise compared against GNN baselines and standard optimization solvers, although experiments are limited.

**Strengths:**

The strengths of the paper can be listed as follows:

1. This work studies an important problem with broad impact, as finding fast solutions to QP problems can benefit many application domains.

2. The direction of integrating diffusion models into learning-to-optimize is interesting, and especially the idea of learning both primal and dual variables jointly through diffusion has not been explored before.

3. The KKT guidance is a smart domain-specific application of energy-based guidance that directly incorporates optimality conditions into the diffusion sampling process.

4. Some promising results are shown comparing against GNNs and standard optimization solvers, though experiments are limited.

**Weaknesses:**

My major concerns for this paper are listed as follows:

1. **Inadequate positioning within the learning-to-optimize literature for QP.** The paper's framing in the abstract, introduction and experiments, focuses almost exclusively on GNN-based end-to-end prediction methods, misleading readers about other relevant baselines in learning-to-optimize for QP. For example, the recent work in [R1] unfolds the ADMM-based OSQP solver (state-of-the-art in QP) [R2] and learns its hyperparameters for solving specific classes of problems faster than classical solvers. The DeepQP framework presented in [R3] (cited in present paper, but the authors only refer to the distributed version) goes further by learning open-loop or closed-loop policies for the parameters to enhance their adaptability across different problems. Another example is [R4], which uses MLP-based prediction followed by fixed OSQP iterations in an end-to-end learning fashion.  The authors should either provide experimental comparisons or substantively discuss how their diffusion-based approach relates to this line of work.

2. **Missing recent work on diffusion-based models for constrained optimization.** The paper claims novelty in applying diffusion models to continuous optimization problems with constraints, but recent work [R5] has already explored diffusion-based models for constrained optimization. The authors should better clarify their contribution relative to existing diffusion-based constrained optimization methods.

3. **Inaccurate claim about achieving optimal solutions.** The claim “Notably, our PDDQP is the first QP neural solver capable of obtaining the optimal solution.” in the abstract and contributions is fundamentally misleading. This doesn’t follow from any guarantees that accompany the diffusion-based model but only from the fact that corrector steps are simply added after the inference of the diffusion model. This post-hoc refinement strategy is standard practice in learning-to-optimize literature, not a novel contribution. See for example, page 2 of [R1] “If run for enough iterations, the iterates are guaranteed to converge to an optimal solution due to the inclusion of the steady-state phase in our architecture.” or then page 6 in the same paper for more details.

4. **Unsupported claim about inferior performance of other generative models.** The authors claim in lines 69-71 and 188-191 that they have “experimented with various models such as self-training consistency models and flow matching, yet their performance is inferior”. However, no experimental results are presented to substantiate this claim. This omission is problematic because understanding why alternative generative approaches fail is essential for evaluating the necessity and contribution of the proposed method.

5. **Limited problems and scale in experiments.** Results are only shown for randomly generated problems that are not inspired by any real-world application. In addition, the problems are of a relatively small scale.

6. The contributions are written in an overly verbose manner, making it hard for the reader to identify the actual intellectual merit of this paper.

Some additional minor comments:

1. The paper has a significant amount of typos or grammar/syntax errors. Examples include lines 13, 87, 124, 128, 132, 157, 188, 243 ("your"), and others.

2. The title and several places in the paper use "primary" instead of "primal" (the standard term in optimization).

3. Section C is the Appendix is empty (no text). At minimum, there should be text pointing to Algorithm 2 or explaining the iterative KKT corrector method.

4. In section D of the Appendix, there is an indicator function for the constraint $Ax=b$ missing in the augmented Lagrangian. In addition, the phrasing “the augmented Lagrangian for the consensus constraint” is also a bit inaccurate as this is the AL for problem (17) in general.

[R1] Sambharya, R., & Stellato, B. (2024). Learning algorithm hyperparameters for fast parametric convex optimization. arXiv preprint arXiv:2411.15717.

[R2] Stellato, Bartolomeo, et al. "OSQP: An operator splitting solver for quadratic programs." Mathematical Programming Computation 12.4 (2020): 637-672.

[R3] Saravanos, Augustinos D., et al. "Deep Distributed Optimization for Large-Scale Quadratic Programming." The Thirteenth International Conference on Learning Representations.

[R4] Sambharya, Rajiv, et al. "Learning to warm-start fixed-point optimization algorithms." Journal of Machine Learning Research 25.166 (2024): 1-46.

[R5] Ding, Shutong, et al. "Exploring the Boundary of Diffusion-based Methods for Solving Constrained Optimization." arXiv preprint arXiv:2502.10330 (2025).

**Questions:**

Some questions for better clarifying the contributions of this work:

1. Can you provide some discussion of what the main novelties of this work are compared to the recent diffusion-based constrained optimization framework in [R5]?

2. Can you provide more extensive experiments on standard QP problems that are relevant to real-world applications (for example, model predictive control, portfolio optimization, etc.)?

3. Can you provide some ablation study on what is the effect of the KKT guidance?

4. What happens when problem structure changes significantly from the training distribution? For example: different sparsity patterns, larger/smaller scale, different constraint/variable ratios, or different conditioning of Q?

5. Can you provide more insights into how much training time does the proposed framework require?

6. Can you provide experimental evidence for the claim that flow Matching and self-trained consistency models fail for QP?

---

### Note · Authors · 2025-11-25

I have read and agree with the venue's withdrawal policy on behalf of myself and my co-authors.